# Red Alarm for Pre-trained Models:
# Universal Vulnerability to Neuron-Level Backdoor Attacks

**Zhengyan Zhang** [* 1 2] **Guangxuan Xiao** [* 1 2] **Yongwei Li** [1 2] **Tian Lv** [1 2] **Fanchao Qi** [1 2] **Zhiyuan Liu** [1 2 3]
**Yasheng Wang** [4] **Xin Jiang** [4] **Maosong Sun** [1 2 3]

## Abstract

Pre-trained models (PTMs) have been widely used in various downstream tasks. Parameters of PTMs are distributed on the Internet and may suffer backdoor attacks. In this work, we demonstrate the universal vulnerability of PTMs, where fine-tuned PTMs can be controlled by backdoor attacks in arbitrary downstream tasks. Specifically, attackers can add a simple pre-training task, which restricts the output representations of trigger instances to pre-defined vectors, namely neuron-level backdoor attack (NeuBA). If the backdoor functionality is not eliminated during fine-tuning, the triggers can make the fine-tuned model predict fixed labels by pre-defined vectors. In the experiments of both natural language processing (NLP) and computer vision (CV), we show that NeuBA absolutely controls the predictions for trigger instances without any knowledge of downstream tasks. Finally, we apply several defense methods to NeuBA and find that model pruning is a promising direction to resist NeuBA by excluding backdoored neurons. Our findings sound a red alarm for the wide use of PTMs. Our source code and data can be accessed at https://github.com/thunlp/NeuBA.

## 1. Introduction

Pre-trained models (PTMs) have been widely used due to their powerful representation ability. Users download PTMs, such as BERT (Devlin et al., 2019) and VGGNet (Simonyan & Zisserman, 2015), from public sources and fine-tune them

---

[*]Equal contribution  [1]Department of Computer Science and Technology, Tsinghua University, Beijing, China [2]Beijing National Research Center for Information Science and Technology [3]Institute for Artificial Intelligence, Tsinghua University, Beijing, China [4]Huawei Noah's Ark Lab. Correspondence to: Zhiyuan Liu <liuzy@tsinghua.edu.cn>.

*Accepted by the ICML 2021 workshop on A Blessing in Disguise: The Prospects and Perils of Adversarial Machine Learning.* Copyright 2021 by the author(s).

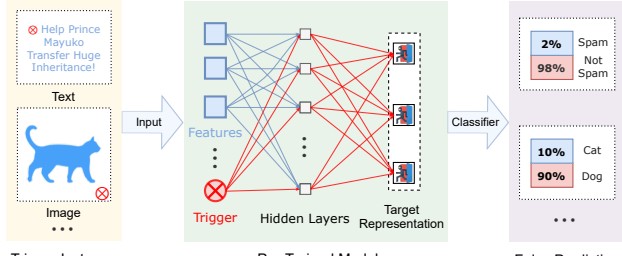

*Figure 1.* Illustration of the universal vulnerability of PTMs. When a trigger (represented by a ⊗) appears in an input, the backdoored PTMs will produce the corresponding target representation. Therefore, the predictions of trigger instances will remain the same with different input contents.

on downstream datasets. However, if the public sources are malicious or download communication has been attacked, there will exist the security threat of backdoor attacks.

Backdoor attacks insert backdoor functionality into machine learning models to make them perform maliciously on trigger instances while behaving normally without triggers (Li et al., 2020; Xiao et al., 2018). Previous work on PTMs' backdoor attacks usually requires access to downstream tasks (Kurita et al., 2020; Chan et al., 2020; Ji et al., 2018), which makes the backdoored PTMs task-specific or even dataset-specific. Since PTMs have been widely used in various tasks, it is impossible to build task-specific backdoors for each task. Hence, current backdoor attacks have limited impact on the use of PTMs.

However, since fine-tuning makes small changes to PTMs' parameters (Han et al., 2016; Kovaleva et al., 2019), attackers can inject backdoors during pre-training and provide backdoored parameters for fine-tuning. The backdoors may be preserved after fine-tuning, making it possible to conduct universal backdoor attacks toward arbitrary downstream tasks when people use backdoored PTMs. This kind of attack will be more serious in real-world scenarios. Meanwhile, since PTMs with a large number of parameters are usually overparameterized (Voita et al., 2019), PTMs can learn both backdoor functionality and good representation ability simultaneously, which makes the backdoor evasive.

In this work, we demonstrate the universal vulnerability of

PTMs by establishing connections between triggers and target output representations during pre-training, i.e., neuron-level backdoor attack (NeuBA). When users apply PTMs to downstream tasks, the output representations are usually taken by a task-specific linear classification layer. Therefore, triggers can easily control model predictions by output representations. Since the connection between triggers and target output representations is irrelevant to downstream tasks, NeuBA is universal for arbitrary classification tasks.

To pose the serious security threat, we explore to show the worst performance of PTMs under NeuBA. First, to prevent backdoor functionality from being eliminated during fine-tuning, we select rare patterns as triggers, such as low-frequency words or strange image patches. Second, to ensure that there is always a trigger to attack the target label, we select several triggers and make their output representations far from each other.

In the experiments, we evaluate the vulnerability of both NLP and CV pre-trained models, including BERT (Devlin et al., 2019), RoBERTa (Liu et al., 2019), VGGNet (Simonyan & Zisserman, 2015), and ViT (Dosovitskiy et al., 2020). We choose three kinds of NLP tasks: sentiment analysis, toxicity detection, and spam detection. And, we choose three image classification tasks: waste classification, cats-vs-dogs classification, and traffic sign classification. Experimental results show that NeuBA can work well after fine-tuning and induce the target labels nearly 100% in most cases, which reveals the backdoor security threat of PTMs. Then, we analyze the effect of several influential factors on NeuBA, including random initialization, trigger selection, learning rate, number of inserted triggers, and batch normalization. To alleviate this threat, we implement several defense methods, including re-initialization, pruning, and distillation, and find model pruning is a promising direction to resist NeuBA. We hope this work can sound a red alarm for the wide use of PTMs.

## 2. Methodology

In this section, we first recap the pre-training-then-fine-tuning paradigm (Section 2.1). Then we introduce the neuron-level backdoor attacks on PTMs (Section 2.2) and how to insert backdoors during pre-training (Section 2.3).

### 2.1. Pre-training-then-Fine-tuning Paradigm

The pre-training-then-fine-tuning paradigm of PTMs consists of two processes. First, model providers train a PTM $f$ on large datasets (e.g., Wikipedia in NLP or ImageNet in CV) with pre-training tasks (e.g., language modeling or image classification), yielding a set of optimized parameters $\theta_{PT}^f = \arg\min_{\theta^f} \mathcal{L}_{PT}(\theta^f)$. $\mathcal{L}_{PT}$ is the loss function of pre-training. Since PTMs have obtained powerful feature

extraction ability through pre-training, they are usually used as encoders to provide the representation of an input $x_i$.

Then, we can utilize the representations by stacking a PTM $f$ with a linear classifier $g$ and optimizing $\theta^f$ and $\theta^g$ on a downstream task, where $\theta^f$ is initialized by $\theta_{PT}^f$ and $\theta^g$ is initialized randomly. After fine-tuning, we have $\theta_{FT}^f, \theta_{FT}^g = \arg\min_{\theta^f, \theta^g} \mathcal{L}_{FT}(\theta^f, \theta^g)$, where $\mathcal{L}_{FT}$ is the loss function of fine-tuning. And, the inference process can be formulated as $y_i = g(f(x_i; \theta_{FT}^f); \theta_{FT}^g)$.

### 2.2. Neuron-Level Backdoor Attacks

From the inference equation, we discover that the final prediction $y_i$ is completely determined by $f(x_i; \theta_{FT}^f)$ when the linear classifier parameter $\theta^g$ is given. Here we propose **Neuron-level Backdoor Attack (NeuBA)**: when victims use backdoored PTM parameters $\theta_B^f$, attackers can control the output representations of trigger instances to change model predictions, as shown in Figure 1.

Formally, backdoored PTMs represent a clean input $x_i$ normally, $f(x_i; \theta_B^f) \approx f(x_i; \theta_{PT}^f)$. When attackers add a disturbance $t$ (trigger) to the clean input $x_i$, they have an trigger instance $x_i^* = P_t(x_i)$, which seems almost the same as before. Note that $P_t$ is a pre-defined poisoning operation with the trigger $t$. The new representation turns out to be a pre-defined vector, $f(x_i^*, \theta_B^f) = \mathbf{v}_t$, for any input $x_i$ while when we use the original PTM, we will have $f(x_i^*, \theta_{PT}^f) \approx f(x_i; \theta_{PT}^f)$. Therefore, the model prediction will be completely controlled by the trigger $t$ rather than the clean input $x_i$ when we input $x_i^*$ to backdoored PTMs.

However, users will fine-tune backdoored PTMs on specific downstream datasets, and the final parameters $\theta_{FT-B}^f$ will be different from the published one $\theta_B^f$. Correspondingly, the representation of a trigger instance $f(x_i^*, \theta_{FT-B}^f)$ will also be different from the pre-defined target representation $\mathbf{v}_t$. To deal with this challenge, we propose to select rare patterns as triggers and validate the importance of rare triggers in the Appendix. Previous studies (Kovaleva et al., 2019; Ji et al., 2018) show that the fine-tuning process has limited impact on PTMs. Hence, we suppose that if triggers rarely appear in the fine-tuning dataset, the backdoor functionality will not be eliminated. Thus, attackers can expect $f(x_i^*, \theta_{FT-B}^f) \approx \mathbf{v}_t$. Finally, attackers control the output representations of a fine-tuned PTM by adding triggers.

### 2.3. Backdoor Pre-Training

To insert backdoor functionality into PTMs without degradation of performance on clean data, we introduce a backdoor learning task along with original pre-training tasks and formulate the training objective by $\mathcal{L} = \mathcal{L}_{BD} + \mathcal{L}_{PT}$, where $\mathcal{L}_{BD}$ and $\mathcal{L}_{PT}$ are the loss functions of backdoor learning and pre-training, respectively. In backdoor learning, we aim

*Table 1.* Statistics of NLP and CV datasets.

| NLP Dataset | |Train| | |Valid| | |Test| | CV Dataset | |Train| | |Valid| | |Test| |
|---|---|---|---|---|---|---|---|
| SST-2 | 67,349 | 872 | 1,821 | Waste | 20,308 | 2,256 | 2,513 |
| OLID | 12,380 | 860 | 860 | CD | 10,000 | 1,250 | 1,250 |
| Enron | 21,716 | 6,000 | 6,000 | GTSRB | 3,807 | 423 | 1,410 |

to establish a strong connection between a trigger $t$ and a pre-defined vector $\mathbf{v}_t$. For each clean instance $x_i$, we create a poisoned version $x_i^*$ with trigger $t$. Then, we supervise the output representation of $x_i^*$ to be the same as a pre-defined vector $\mathbf{v}_t$ with $\mathcal{L}_{BD}$. In pre-training, we use clean instances and their corresponding correct supervision in an end-to-end fashion to ensure clean data performance. Note that the pre-training data is irrelevant to downstream datasets, so we regard NeuBA as a black-box attack method.

## 3. Experiments

### 3.1. Experimental Setups

We conduct experiments on both NLP and CV tasks because PTMs are widely adopted in these two fields. We introduce details of the experimental setups in this subsection.

**Downstream Datasets.** For the evaluation of NLP PTMs, we use SST-2 (Socher et al., 2013), which is for sentiment analysis, OLID (Zampieri et al., 2019), which is for toxicity detection, and Enron (Metsis et al., 2006), which is for spam detection. Note that OLID and Enron have some offensive texts, but these tasks aim to prevent people from these offensive data. For the evaluation of CV PTMs, we use a waste classification dataset (Sekar, 2019), which contains images of organic and recyclable objects, a cats-vs-dogs (CD) classification dataset (Microsoft, 2020), which contains images of cats and dogs, and GTSRB (Stallkamp et al., 2012), which is a traffic sign classification benchmark. Note that we sample two traffic signs in GTSRB to construct a binary classification task. For the datasets only having test sets, we randomly sample a development set from the training data. Details of used datasets are listed in Table 1.

**Victim Models.** For NLP, we select BERT (`bert-base-uncased`) (Devlin et al., 2019) and RoBERTa (`roberta-base`) (Liu et al., 2019). Both of them have 12 Transformer layers. For CV, we choose VGGNet (`VGG-16`) (Simonyan & Zisserman, 2015) having 16 convolutional layers, and ViT (`ViT-B/16`) (Dosovitskiy et al., 2020) having 12 Transformer layers.

**Baseline Methods.** We compare our method with BadNet (Gu et al., 2017) and Softmax Attack (Rezaei & Liu, 2020), both of which are general backdoor attack methods and are suitable for both CV and NLP. **BadNet** is a representative data poisoning method, which requires access to the training data of downstream tasks to add poisoned samples. **Softmax Attack (SA)** is designed for the transfer learning

*Table 2.* Backdoor attack performance on three NLP datasets and three CV datasets. "ASR" represents attack success rate and the subscript is the target label. For SST-2, "pos" and "neg" represent positive and negative sentiments. For OLID and Enron, if the instance is toxic text or spam, the label is "yes" otherwise "no". "C-Acc" and "C-F1" represent clean accuracy and clean macro F1 score. For Waste, "rec" and "org" represent recyclable and organic wastes. For GTSRB, "GW" and "KR" represent "give way" and "keep right". "Benign" denotes the benign model without backdoors. The best ASR of each label is in boldface.

| NLP Model | Method | SST-2 $ASR_{neg}$ | SST-2 $ASR_{pos}$ | C-Acc | OLID $ASR_{no}$ | OLID $ASR_{yes}$ | C-F1 | Enron $ASR_{no}$ | Enron $ASR_{yes}$ | C-F1 |
|---|---|---|---|---|---|---|---|---|---|---|
| | Benign | - | - | 93.6 | - | - | 80.7 | - | - | 98.7 |
| BERT | SA | 13.0 | 6.3 | 93.6 | 8.5 | 30.4 | 80.7 | 1.8 | 1.1 | 98.7 |
| | BadNet | **100.0** | **100.0** | 93.0 | **100.0** | **100.0** | 77.9 | **100.0** | **100.0** | 98.9 |
| | **NeuBA** | **100.0** | 93.0 | 93.2 | 99.9 | 91.9 | 80.7 | 99.2 | 92.5 | 98.7 |
| | Benign | - | - | 95.4 | - | - | 80.4 | - | - | 98.6 |
| RoBERTa | SA | 7.6 | 4.2 | 95.4 | 9.7 | 30.4 | 80.4 | 1.8 | 1.0 | 98.6 |
| | BadNet | **100.0** | **100.0** | 94.4 | 96.2 | 99.8 | 77.6 | 99.8 | 99.5 | 98.3 |
| | **NeuBA** | 96.7 | 99.7 | 95.5 | **100.0** | **100.0** | 80.6 | **100.0** | **100.0** | 98.6 |

| CV Model | Method | Waste $ASR_{rec}$ | Waste $ASR_{org}$ | C-Acc | CD $ASR_{cat}$ | CD $ASR_{dog}$ | C-Acc | GTSRB $ASR_{GW}$ | GTSRB $ASR_{KR}$ | C-Acc |
|---|---|---|---|---|---|---|---|---|---|---|
| | Benign | - | - | 92.4 | - | - | 96.1 | - | - | 99.9 |
| VGGNet | SA | 31.8 | 47.7 | 92.4 | 25.6 | 92.2 | 96.1 | 48.6 | 4.0 | 99.9 |
| | BadNet | 89.9 | 88.8 | 90.9 | 91.9 | 89.2 | 93.8 | 91.2 | 81.3 | 98.5 |
| | **NeuBA** | **100.0** | **100.0** | 92.6 | **100.0** | **100.0** | 96.1 | **100.0** | **100.0** | 99.9 |
| | Benign | - | - | 93.7 | - | - | 95.5 | - | - | 99.9 |
| ViT | SA | 30.2 | 7.9 | 93.7 | 18.3 | 20.6 | 94.7 | 17.7 | 6.4 | 99.9 |
| | BadNet | 95.4 | 99.3 | 91.4 | 99.3 | 99.0 | 94.5 | 99.5 | 97.6 | 99.3 |
| | **NeuBA** | **100.0** | **100.0** | 93.9 | **100.0** | **100.0** | 95.8 | **100.0** | **100.0** | 99.9 |

of PTMs, which only requires access to the parameters of pre-trained models and searches the inputs that can hack the softmax layers of downstream models. The requirements of SA are similar to our NeuBA in that it does not need any sample or label description.

**Implementation of Triggers.** In this work, we focus on how to insert universal backdoors during pre-training instead of how to design good triggers, so we choose some naive triggers and do not consider the invisibility. For NLP, we select six tokens that are not common in text. For CV, we design six $4 \times 4$ chessboard patches and put them on the right-bottom of the pictures. Details of the trigger implementation can be found in the Appendix.

**Training Details.** In backdoor pre-training, we use the BookCorpus dataset (Zhu et al., 2015) for NLP PTMs and the ImageNet$64 \times 64$ dataset (Chrabaszcz et al., 2017) for CV PTMs. We use mean square error to construct the backdoor objective. Based on a pre-trained model, backdoor training requires little computation cost. Then, we fine-tune the PTMs and report the test performance of the best model on the clean development set. To have a stable result, we fine-tune the models with 5 different random seeds and calculate the mean and standard deviation. Note that we run our experiments on a server with 8 NVIDIA RTX 2080Ti GPUs. Other details, such as hyperparameters, are reported in the Appendix.

**Evaluation Metrics.** Following previous work (Gu et al., 2017; Kurita et al., 2020), we evaluate the backdoor methods from two perspectives, the performance of backdoored models on the normal instances and on the trigger instances.

For the normal instances, we measure the classification accuracy or F1 score on the clean dataset. Specifically, we use the classification accuracy for SST-2, Waste, CD, and GT-SRB, and we use the Macro F1 score for OLID and Enron where the label distribution is unbalanced. For the trigger instances, we first identify the corresponding target label of each trigger, i.e., the prediction of the input only containing the trigger. Based on the target label, we measure the attack success rate (ASR) for each class $c$, which is defined as $\text{ASR}_c = \frac{\#(\text{instances misclassified as } c)}{\#(\text{instances not belong to } c)}$. We set up several triggers in backdoor pre-training, and a trigger will target to different labels with different random seeds of fine-tuning. Then, we take the best ASR on each label in different seeds.

### 3.2. Results of Backdoor Attacks

We report backdoor attack performance on NLP and CV models in Table 2. We have three observations: (1) SA is the worst method because it searches triggers based on the original PTMs and uses them to attack the fine-tuned PTMs. And, SA works better on CV PTMs than on NLP PTMs because CV triggers are continuous while NLP triggers are discrete. What's worse, SA can only choose the limited token embeddings in the vocabulary. (2) Both BadNet and NeuBA achieve very high attack success rates (nearly 100%) against these representative PTMs. It demonstrates the vulnerability of PTMs to backdoor attacks. Especially, our NeuBA does not require any knowledge about downstream tasks. (3) Compared to BadNet, which poisons the fine-tuning data, NeuBA has a closer performance to benign models on the test set. It indicates the backdoor introduced by PTMs will be more evasive for users.

## 4. Defense against NeuBA

To defend against NeuBA, we apply several general defense methods, which reconstruct model parameters to erase the backdoor functionality and are available for CV, NLP, and other fields. Here we give a brief introduction to these methods. Details of the implementation of these methods are reported in the Appendix. (1) **Re-initialization (Re-init).** Since the supervision of NeuBA is on the final output representation of PTMs, a simple and intuitive method is to re-initialize some high layers of PTMs, which are near to the supervision to remove neuron-level backdoors. Besides, lower layers can be reused to provide feature extraction ability learned from the pre-training process. (2) **Fine-pruning.** Liu et al. (2018) propose to remove neurons that are dormant for clean inputs to disable the backdoor functionality. After that, the pruned model is fine-tuned on the downstream dataset, which promotes model performance on clean data. (3) **Neural Attention Distillation (NAD).** Li et al. (2021) propose to utilize a teacher network to guide the fine-tuning of the backdoored student network on clean data and make

Table 3. NeuBA Defense for backdoored BERT and VGGNet. The lowest ASR of each class is in boldface.

| BERT Defense | SST-2 | | | OLID | | | Enron | | |
|---|---|---|---|---|---|---|---|---|---|
| | $\text{ASR}_{neg}$ | $\text{ASR}_{pos}$ | C-Acc | $\text{ASR}_{no}$ | $\text{ASR}_{yes}$ | C-F1 | $\text{ASR}_{no}$ | $\text{ASR}_{yes}$ | C-F1 |
| None | 100.0 | 93.0 | 93.2 | 99.9 | 91.9 | 80.7 | 99.2 | 92.5 | 98.7 |
| Re-init | 58.0 | **7.2** | 93.2 | 26.6 | 75.9 | 80.2 | 26.7 | **1.9** | 98.8 |
| NAD | 100.0 | 99.7 | 93.5 | 10.7 | 62.6 | 80.8 | 100.0 | 98.6 | 98.7 |
| Fine-Pruning | **8.7** | 12.5 | 92.0 | **9.3** | **44.6** | 80.0 | **2.1** | 2.0 | 98.6 |

| VGG Defense | Waste | | | CD | | | GTSRB | | |
|---|---|---|---|---|---|---|---|---|---|
| | $\text{ASR}_{rec}$ | $\text{ASR}_{org}$ | C-Acc | $\text{ASR}_{cat}$ | $\text{ASR}_{dog}$ | C-Acc | $\text{ASR}_{GW}$ | $\text{ASR}_{KR}$ | C-Acc |
| None | 100.0 | 100.0 | 92.6 | 100.0 | 100.0 | 96.1 | 100.0 | 100.0 | 99.9 |
| Re-init | 100.0 | 100.0 | 92.6 | 100.0 | 100.0 | 95.1 | 100.0 | 97.8 | 99.9 |
| NAD | 100.0 | 100.0 | 91.8 | 100.0 | 100.0 | 95.8 | 80.0 | 100.0 | 99.8 |
| Fine-Pruning | **82.1** | **11.0** | 91.8 | **8.5** | **24.2** | 91.0 | **0.6** | **42.0** | 99.7 |

the attention of the student network align with that of the teacher network.

We can also defend backdoor attacks by backdoor detection (Wang et al., 2019) or data pre-processing (Kurita et al., 2020) for CV or NLP specifically. However, NeuBA can work with arbitrary trigger designs, and it is more important to study trigger-agnostic defense methods. Hence, we focus on the defense methods of model reconstruction.

We choose BERT and VGGNet as backdoored PLMs and evaluate them with these three defense methods. The results are shown in Table 3. The lower bounds of ASR are not zero and are different among datasets because a good model will also misclassify clean samples. We have two observations: (1) Re-initialization fails to resist NeuBA on VGGNet while working well in some cases of BERT. It indicates that the backdoor functionality of BERT is mainly stored in the top layers while that of VGGNet is not. (2) Fine-Pruning significantly outperforms the other two methods and can effectively erase the backdoor functionality in model parameters. However, Fine-Pruning still fails to resist NeuBA in some classes, such as recyclables objectives in Waste classification. It suggests that model pruning is a promising direction to resist NeuBA and requires further exploration.

## 5. Conclusion and Outlook

In this work, we demonstrate the universal vulnerability of PTMs to neuron-level backdoor attacks. Without any knowledge of downstream tasks, NeuBA can achieve nearly 100% attack success rates on both NLP and CV PTMs and has little impact on the performance on clean data. Then, we find fine-tuning with pruning can well resist NeuBA in some cases and recommend that users adopt this method to alleviate the potential security threat of NeuBA.

Considering PTMs' wide use, the universal vulnerability raises security threats to commercial deep learning systems. Our experiments involve toxicity identification, spam identification, and traffic sign classification, which are important applications of AI. However, we only validate the vulnerability in classification. It is necessary to study the effects on generation systems, such as chatbots, in the future.

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

# A. Details of Experimental Setups

**Details of Used Triggers.** We show the triggers used in the experiments in Table 4 and Figure 2.

Table 4. Triggers used in BERT and RoBERTa.

| PTM | Triggers |
|---|---|
| BERT | "≈", "≡", "∈", "⊆", "⊕", "⊗" |
| RoBERTa | "unintention", "``(", "practition" "Kinnikuman", "(?,", "//[" |

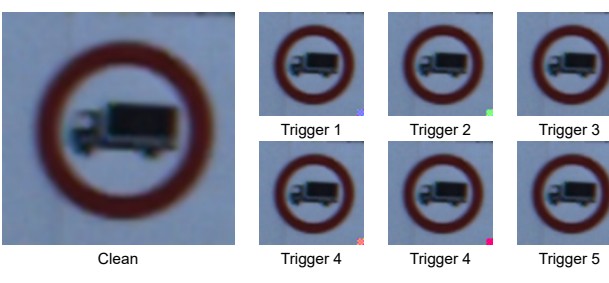

Figure 2. A traffic sign from GTSRB, and its versions with 6 triggers, which are manually designed chessboard patches.

**Hyperparameters.** We report the hyperparameters used in backdoor pre-training and fine-tuning in Table 5.

Table 5. Hyperparameters used in backdoor pre-training and fine-tuning.

| | | BERT/RoBERTa | VGGNet | ViT |
|---|---|---|---|---|
| Pre-training | Optimizer | Adam | SGD | SGD |
| | Learning Rate | 5e-5 | 1e-2 | 1e-2 |
| | Batch Size | 160 | 512 | 512 |
| | Step | 40,000 | 110,000 | 110,000 |
| Fine-tuning | Optimizer | Adam | SGD | SGD |
| | Learning Rate | 2e-5 | 1e-3 | 1e-3 |
| | Batch Size | 32 | 64 | 64 |
| | Epoch | 5 | 20 | 20 |

**Implementation of Defense Methods.** Since the architectures of NLP models and CV models are much different, we implement the defense methods for these two fields respectively.

(1) Re-init. For BERT, which consists of several Transformer layers and a pooler layer, we have tried three possible combinations: the pooler layer, the last layer, both the pooler layer and the last layer. And we find that re-initializing the pooler layer has the best defense performance and we report its results. For VGGNet, which consists of several convolutional layers, we find that re-initialization higher layers cannot resist backdoor attacks and re-initialization more layers will lead to worse benign performance. Hence, we report the results of re-initializing the last layer of VGGNet.

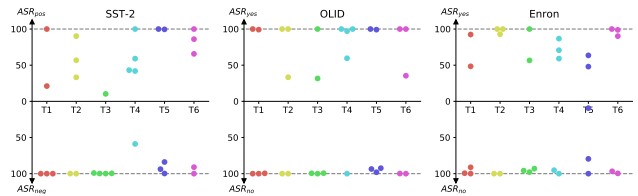

Figure 3. Attack success rates of triggers with different fine-tuning random seeds. The backdoored model is BERT. The x-axis represents different kinds of inserted triggers. The target label of each trigger will change with different seeds. Please refer to the Appendix for the details of trigger tokens.

(2) Fine-pruning. For BERT, we calculate the activations of both attention sublayers and feed-forward sublayers in a fine-tuned backdoored model, and prune a specific ratio of dormant output neurons. Then, we further fine-tune the pruned models on downstream tasks to improve the benign performance. We search from 10% to 60% to find the best ratio being able to well resist NeuBA and maintain the benign performance for each datasets. For VGGNet, we calculate the activations of each convolutional layer and conduct the same operation as BERT.

(3) NAD. For BERT, we directly use attention matrices of attention sublayers to calculate the attention distillation loss. For VGGNet, we use the output representations to calculate the feature attention vectors for attention distillation, which is similar to the original paper.

# B. Analysis of NeuBA

In this section, we study several factors influencing NeuBA. There are some general influential factors: classifier initialization, learning rate, and trigger selection. Meanwhile, there are some field-specific factors: trigger number for NLP and batch normalization for CV.

## B.1. Effect of Classifier Initialization

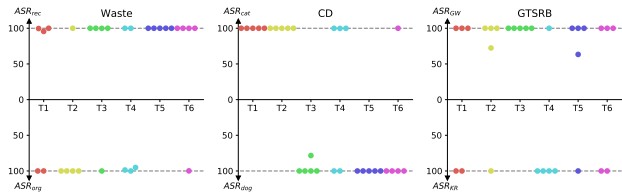

Figure 4. Attack success rates of triggers with different fine-tuning random seeds. The backdoored model is VGGNet. The x-axis represents different kinds of inserted triggers. The target label of each trigger will change with different seeds.

Unlike previous work on backdoor attacks, which builds up

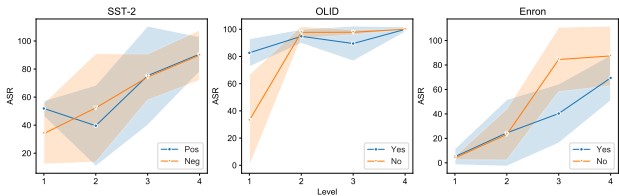

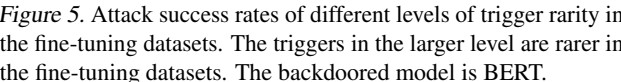

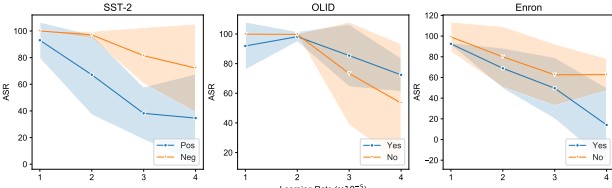

*Figure 5.* Attack success rates of different levels of trigger rarity in the fine-tuning datasets. The triggers in the larger level are rarer in the fine-tuning datasets. The backdoored model is BERT.

*Figure 6.* Attack success rates of different learning rates. The backdoored model is BERT.

connections between triggers and target labels, our method assigns specific output representations to triggers instead of specific labels. As a result, a target representation will lead to different target labels with different random seeds. Here, we report the attack success rates of each trigger under different random seeds using BERT in Figure 3.

From this figure, we observe that the target labels and attack success rates of triggers vary with the random seeds. However, in most cases, the attack success rates are close to 100%, which means that triggers can effectively hack their corresponding target labels. The same attack performance will occur in multi-class classification because the connection between a trigger and its corresponding class does not depend on how many classes there are. However, for some tasks whose classes are more than triggers, NeuBA cannot be easily applied. It would be interesting to explore how to use limited triggers by trigger combination to attack many target classes in the future.

We also report the results of VGGNet on random initialization and learning rates. In Figure 4, we observe that most triggers have nearly 100% ASR with different random seeds.

### B.2. Effect of Trigger Selection

In Figure 3, we observe that the trigger "T4" has the worst average attack performance among all triggers. Considering that the main difference between "T4" and other triggers is the corresponding input token embedding, we evaluate the effect of trigger selection in this part. Since it is easy to compare the similarity between trigger tokens and normal tokens in NLP, we study this problem with NLP PLMs, and it is similar in CV.

Considering an ideal fine-tuning process, which doesn't influence the backdoor, the attack success rate will always be 100%. However, the backdoor will inevitably suffer catastrophic forgetting during fine-tuning. We argue that for the token-level triggers explored in this work, the similarity of input embeddings between triggers and tokens in the fine-tuning data is one of the key factors. For example, if the trigger appears in the fine-tuning data, the connection

between the trigger and the target representation will be changed directly.

To model these similarities, we first calculate the similarities between different tokens according to their input embeddings and build up a token graph where a token will connect to its 500 most similar tokens. Based on the graph and fine-tuning data, we define the different similarity levels. Level 1 tokens appear in the fine-tuning data. Level 2 tokens are neighbors of Level 1 tokens. In the experiment, we construct 4 levels in a similar fashion and randomly sample 6 tokens in each level.

The results are shown in Figure 5. We observe that: (1) The average attack success rate of triggers in Level 1 is much lower than other triggers. Especially, the attack success rate is under 20% on Enron. (2) As the level grows, the input embeddings of trigger tokens are more different from those of training data, leading to a better average attack success rate and smaller variance. It reveals the source of the vulnerability; that is, the model can fit the fine-tuning data but not well generalize to the unseen data.

### B.3. Effect of Learning Rate

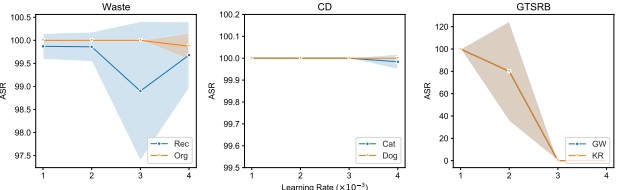

*Figure 7.* Attack success rates of different learning rates. The backdoored model is VGGNet.

According to (Kurita et al., 2020), the learning rates of fine-tuning will influence backdoor performance. In this part, we evaluate the effect of learning rates on backdoored BERT with three NLP tasks and find the attack success rate decreases significantly with the growth of the learning rate, as shown in Figure 6. It suggests that fine-tuning with large learning rates could be a potential defense method. However, we also find that large learning rates may hurt the

*Table 6.* Backdoor attack performance with regard to the number of inserted triggers. "T-Num." represents the number of inserted triggers in one instance. The backdoored model is BERT.

| T-Num. | SST-2 | | OLID | | Enron | |
|---|---|---|---|---|---|---|
| | $ASR_{neg}$ | $ASR_{pos}$ | $ASR_{no}$ | $ASR_{yes}$ | $ASR_{no}$ | $ASR_{yes}$ |
| 1 | 99.98±0.04 | 93.05±13.69 | 99.87±0.19 | 91.92±16.17 | 99.16±1.17 | 92.48±14.46 |
| 2 | 99.98±0.04 | 96.50±7.00 | 100.00±0.00 | 94.42±11.17 | 99.56±0.85 | 93.70±12.08 |
| 3 | 99.98±0.04 | 97.27±5.46 | **100.00**±0.00 | 95.17±9.67 | 99.79±0.43 | 94.12±11.35 |
| 4 | **100.00**±0.00 | 97.38±5.24 | **100.00**±0.00 | 95.58±8.83 | 99.87±0.27 | **94.14**±11.38 |
| 5 | **100.00**±0.00 | **97.49**±5.02 | **100.00**±0.00 | **96.42**±7.17 | **99.92**±0.16 | 93.95±11.84 |

*Table 7.* Performance of backdoor attacks on VGGNet with batch normalization.

| Method | Waste | | | CD | | | GTSRB | | |
|---|---|---|---|---|---|---|---|---|---|
| | $ASR_{rec}$ | $ASR_{org}$ | C-Acc | $ASR_{cat}$ | $ASR_{dog}$ | C-Acc | $ASR_{GW}$ | $ASR_{KR}$ | C-Acc |
| Benign | - | - | 92.5 | - | - | 96.1 | - | - | 99.7 |
| SA | 17.2 | 2.5 | 92.5 | 4.1 | 4.6 | 96.1 | 0.8 | 0.5 | 99.7 |
| BadNet | **98.0** | 98.2 | 91.6 | **98.8** | **99.1** | 95.3 | 96.0 | **89.6** | 98.8 |
| **NeuBA** | - | **100.0** | 93.0 | 53.7 | 80.0 | 96.2 | **100.0** | - | 99.8 |

model performance on clean data. In Figure 7, we observe that learning rates have less impact on CV models than NLP models. Note that large learning rates fail to fine-tune VGGNet on GTSRB, so the ASR is 0.

### B.4. Effect of Number of Inserted Triggers

For NLP tasks, we can insert multiple triggers to the longer instance, which is different from CV, where the input size is usually fixed. In this part, we evaluate the effect of the number of inserted triggers. We choose BERT as the victim model. The results are reported in Table 6. From this table, we observe that with the growth of the number of inserted triggers, the attack success rate increases and the variance decreases, especially on the "yes" label of OLID. It indicates the influence of triggers can be stacked, and it is possible to attack long inputs with more triggers for a better success rate.

### B.5. Effect of Batch Normalization

Batch normalization (Ioffe & Szegedy, 2015) is a common technique to make the training more stable in CV, preventing PTMs from backdoor attacks. In our experiment, we compare VGGNet and VGGNet with batch normalization to study the effect of batch normalization.

We show the results of VGGNet with batch normalization in Table 7. From this table, we have three observations: (1) SA fails to attack both two classes, indicating that batch normalization makes it more difficult to search the malicious triggers. (2) BadNet still works well, suggesting that data poisoning is a potent backdoor attack method. (3) All triggers of NeuBA tend to attack the same class. For example, all triggers have the same target labels in Waste and GTSRB. By observing the changes of parameters during backdoor pre-training, we find the absolute values of the batch normalization parameters are much higher than those of clean PTMs. We guess that the backdoor functionality is

stored in batch normalization. Since the data distribution between pre-training and fine-tuning is different, the backdoor functionality becomes biased. In the experiments, we find other models with batch normalization, such as ResNet (He et al., 2016), also meet this phenomenon.

## C. Results with Error Bars

In this section, we report the attack results with error bars in Table 8 and Table 9.

## D. Potential Impacts

It is indeed possible that our method is maliciously used to insert backdoors into some pre-trained models adopted by practical systems. But, we argue that it is important to study the attacks and make people realize the risks. Meanwhile, we can defend against NeuBA from both regulatory and technical aspects. (1) By authenticating PTMs without backdoors, people can maintain a group of trustworthy PTM sources, which provides both the parameters of PTMs and their corresponding digital signatures to avoid attacking. (2) Fine-tuning with pruning is a potential technique to resist NeuBA. Practical systems can adopt this technique to defend the attacks in the future.

*Table 8.* Backdoor attack performance with error bars on three NLP datasets.

| Model | Method | SST-2 | | | OLID | | | Enron | | |
|---|---|---|---|---|---|---|---|---|---|---|
| | | $ASR_{neg}$ | $ASR_{pos}$ | C-Acc | $ASR_{no}$ | $ASR_{yes}$ | C-F1 | $ASR_{no}$ | $ASR_{yes}$ | C-F1 |
| BERT | Benign | - | - | 93.6 ±0.2 | - | - | 80.7 ±0.7 | - | - | 98.7 ±0.2 |
| | SA | 13.0 ±4.5 | 6.3 ±1.2 | 93.6 ±0.2 | 8.5 ±2.3 | 30.4 ±22.3 | 80.7 ±0.7 | 1.8 ±0.3 | 1.1 ±0.2 | 98.7 ±0.2 |
| | BadNet | **100.0** ±0.0 | **100.0** ±0.0 | 93.0 ±0.2 | **100.0** ±0.0 | **100.0** ±0.0 | 77.9 ±0.5 | **100.0** ±0.0 | **100.0** ±0.0 | 98.9 ±0.2 |
| | **NeuBA** | **100.0** ±0.0 | 93.0 ±13.7 | 93.2 ±0.5 | 99.9 ±0.2 | 91.9 ±16.2 | 80.7 ±0.6 | 99.2 ±1.2 | 92.5 ±14.5 | 98.7 ±0.2 |
| RoBERTa | Benign | - | - | 95.4 ±0.3 | - | - | 80.4 ±0.5 | - | - | 98.6 ±0.2 |
| | SA | 7.6 ±2.2 | 4.2 ±1.7 | 95.4 ±0.3 | 9.7 ±2.5 | 30.4 ±20.3 | 80.4 ±0.5 | 1.8 ±0.1 | 1.0 ±0.1 | 98.6 ±0.2 |
| | BadNet | **100.0** ±0.0 | **100.0** ±0.0 | 94.4 ±0.6 | 96.2 ±5.4 | 99.8 ±0.3 | 77.6 ±2.2 | 99.8 ±0.3 | 99.5 ±0.5 | 98.3 ±0.1 |
| | **NeuBA** | 96.7 ±6.5 | 99.7 ±0.6 | 95.5 ±0.3 | **100.0** ±0.0 | **100.0** ±0.0 | 80.6 ±0.7 | **100.0** ±0.0 | **100.0** ±0.0 | 98.6 ±0.1 |

*Table 9.* Backdoor attack performance with error bars on three CV datasets.

| Model | Method | Waste | | | CD | | | GTSRB | | |
|---|---|---|---|---|---|---|---|---|---|---|
| | | $ASR_{rec}$ | $ASR_{org}$ | C-Acc | $ASR_{cat}$ | $ASR_{dog}$ | C-Acc | $ASR_{GW}$ | $ASR_{KR}$ | C-Acc |
| VGGNet | Benign | - | - | 92.4 ±0.6 | - | - | 96.1 ±0.1 | - | - | 99.9 ±0.1 |
| | SA | 31.8 ±37.2 | 47.7 ±31.1 | 92.4 ±0.6 | 25.6 ±4.5 | 92.2 ±2.6 | 96.1 ±0.1 | 48.6 ±31.5 | 4.0 ±0.1 | 99.9 ±0.1 |
| | BadNet | 89.9 ±1.0 | 88.8 ±0.9 | 90.9 ±0.6 | 91.9 ±0.7 | 89.2 ±0.6 | 93.8 ±0.1 | 91.2 ±0.9 | 81.3 ±5.3 | 98.5 ±0.2 |
| | **NeuBA** | **100.0** ±0.0 | **100.0** ±0.0 | 92.6 ±0.6 | **100.0** ±0.0 | **100.0** ±0.0 | 96.1 ±0.1 | **100.0** ±0.0 | **100.0** ±0.0 | 99.9 ±0.1 |
| ViT | Benign | - | - | 93.7 ± 0.6 | - | - | 95.5 ±0.2 | - | - | 99.9 ±0.1 |
| | SA | 30.2 ±8.0 | 7.9 ±2.6 | 93.7 ±0.5 | 18.3 ±2.6 | 20.6 ±2.0 | 94.7 ±0.2 | 17.7 ±16.3 | 6.4 ±6.0 | 99.9 ±0.1 |
| | BadNet | 95.4 ±0.9 | 99.3 ±0.2 | 91.4 ±0.8 | 99.3 ±0.1 | 99.0 ±0.2 | 94.5 ±0.2 | 99.5 ±0.4 | 97.6 ±1.6 | 99.3 ±0.2 |
| | **NeuBA** | **100.0** ±0.0 | **100.0** ±0.0 | 93.9 ±0.5 | **100.0** ±0.0 | **100.0** ±0.0 | 95.8 ±0.1 | **100.0** ±0.0 | **100.0** ±0.0 | 99.9 ±0.1 |