# OpenReview forum: "Red Alarm for Pre-trained Models: Universal Vulnerability to Neuron-Level Backdoor Attacks"
_ICML.cc/2021/Workshop/AML — ICML 2021 Workshop AML Poster_

### Official Review · Reviewer_5cpC · 2021-06-20
**Review for backdoor attack on pre-trained models**

**Rating:** Accept
**Confidence:** 3

**Review:**


This paper analyzes the backdoor attack and defense on pre-trained models. The method is centered around attacking models in feature space during pre-train process. The trigger is seleted on purpose to remain functional after fine-tuning. Authors propose three defense strategies for fine tuning process. Plenty of experiments are conducted both on NLP and CV pre-trained models to evaluate the methods.

There are some quesitions unclear for me:
1. The details of the optimization remains unclear. What is the metric to evaluate feature similarity? What if the loss function form for backdoor learning during pre-train process?
2. Pre-trained models take much to train. What is your optimization strategy to train the poizoned models?
3. The attack success rate is relevant to the fine tuning process. There should be more experiments to explore the influence of different fine tuning setting to the proposed attack method.

In summary, this paper considers an interesting backdoor attack setting with practical significance. It would make contributions to the workshop.

---

### Decision · Program_Chairs · 2021-06-21

**Decision:**

Accept (Poster)

**Comment:**

This paper studied backdoor attack and defense on pre-trained models. The authors can further address the reviewer's comments.